# Colloidal Behavior and Biodegradation of Engineered Carbon-Based Nanomaterials in Aquatic Environment

**DOI:** 10.3390/nano12234149

**Published:** 2022-11-23

**Authors:** Konstantin Pikula, Seyed Ali Johari, Kirill Golokhvast

**Affiliations:** 1Polytechnical Institute, Far Eastern Federal University, 10 Ajax Bay, Russky Island, Vladivostok 690922, Russia; 2Department of Fisheries, Faculty of Natural Resources, University of Kurdistan, Pasdaran St., Sanandaj 66177-15175, Iran; 3Siberian Federal Scientific Centre of Agrobiotechnology, Centralnaya, Presidium, Krasnoobsk 633501, Russia

**Keywords:** biodegradation, carbon dots, enzymes, fate, fullerenes, graphene, nanotubes, aquatic species

## Abstract

Carbon-based nanomaterials (CNMs) have attracted a growing interest over the last decades. They have become a material commonly used in industry, consumer products, water purification, and medicine. Despite this, the safety and toxic properties of different types of CNMs are still debatable. Multiple studies in recent years highlight the toxicity of CNMs in relation to aquatic organisms, including bacteria, microalgae, bivalves, sea urchins, and other species. However, the aspects that have significant influence on the toxic properties of CNMs in the aquatic environment are often not considered in research works and require further study. In this work, we summarized the current knowledge of colloidal behavior, transformation, and biodegradation of different types of CNMs, including graphene and graphene-related materials, carbon nanotubes, fullerenes, and carbon quantum dots. The other part of this work represents an overview of the known mechanisms of CNMs’ biodegradation and discusses current research works relating to the biodegradation of CNMs in aquatic species. The knowledge about the biodegradation of nanomaterials will facilitate the development of the principals of “biodegradable-by-design” nanoparticles which have promising application in medicine as nano-carriers and represent lower toxicity and risks for living species and the environment.

## 1. Introduction

The production of nanomaterials has become a multibillion-dollar industry, and nanoparticles (NPs) are currently used in various consumer products, such as sprays, powders, food additives, sunscreens, medicines, etc. [1]. At the same time, there is a great concern regarding the negative impact and threat of NPs to the environment and human health [2,3,4]. The regulation of environmental and human safety of nanoproducts is challenged by a lack of knowledge about their toxic behavior [5]. This problem is made up of multiple combinations of nanomaterial properties, which affect their toxicity, such as type, size, shape, impurity, agglomeration, degradation properties, and accumulation in flora and fauna [1].

According to the composition classification, the NPs can be divided in four categories, namely (1) carbon-based, (2) inorganic-based, (3) organic-based, and (4) composite-based [6]. The following discussion in this review will be focused on the group of carbon-based nanomaterials (CNMs), which have achieved exponentially growing interest over the last decades [7,8] due to their superb electrical and heat conductivity, optical properties, mechanical strength, chemical stability, and other advantageous physical and chemical properties [9].

Rapid growth of the CNMs market has been catalyzed by the increased number of applications, including the automotive and aerospace industry, lithium-ion batteries, medicine and pharmaceuticals, electrochemistry, catalyst production, and pollutant removal [10]. The increasing volume of CNMs’ production has led to higher opportunities of environmental contamination and it can be a serious threat to living organisms. The impact of CNMs on aquatic organisms represents significant interest because of their inevitable contact with the aquatic environment.

One of the important sources of aquatic contamination is the application of CNMs for water purification as a cost-effective and eco-friendly option [11,12,13]. Moreover, CNMs are widely used in agriculture [14] which as the other source of water contamination by further surface wash. The largest group of contamination sources is consumer products containing CNMs, which could release NPs at each stage of the life cycle [15]. The released CNMs finally occur in water bodies with precipitation, sedimentation, and surface wash [16]. Multiple studies in recent years highlight the toxicity of CNMs in relation to aquatic organisms, including bacteria [17], microalgae [18,19], bivalves [20,21], sea urchins [20,22], and other species. Therefore, the fate and toxicity of CNMs in the aquatic environment has required careful study.

As we mentioned above, the toxicity of NPs depends on a combination of multiple factors and properties of the particles. In this regard, CNMs should be grouped into classes. Based on dimensional classification, CNMs have groups: 0D, 1D, 2D, and 3D, where the number before the D means the number of dimensions of NPs in size above 100 nm. The group 0D includes fullerenes, onion-like carbon, carbon dots, and nanodiamonds. The particles of the 1D group includes nanofibers, nanotubes, and nanohorns. The group 2D includes multilayer graphitic nanosheets, graphene nanoribbons, and graphene-related materials [15,23].

Considering the significant difference in physical and chemical properties, all these types of CNMs will have different toxic behavior. Moreover, in the aquatic environment, the factor of NPs’ transformation obtains essential meaning, and it can further differ the initial toxic properties of the substance [24]. Therefore, the fate and behavior of CNMs in the aquatic environment plays an important role in understanding the possible influence on the aquatic environment.

The initial idea of this review was to highlight the fate of CNMs in the aquatic environment and to consider the possibility of aquatic species to degrade synthetic CNMs. This opportunity might be obtained by some aquatic species after a long historical period of contact with natural and anthropogenic carbon NPs, such as soot, fossil coal, wildfire charcoal, carbon black, fuel combustion, etc. [25,26].

Work with bibliographic sources was supported from the State Assignment of the Ministry of Science and Higher Education of the Russian Federation №0657-2020-0013.

## 2. Colloidal Behavior and Stability of Carbon-Based Nanomaterials in the Aquatic Environment

### 2.1. The Main Principles of Nanomaterial Transformation in the Aquatic Environment

The duration of NMs’ persistence in aquatic systems remains an ongoing concern regarding potential long-term toxicity. Therefore, an understanding of their transformation and degradation in an aquatic environment is an important issue to maintain the safe utilization of NMs.

After entering the aquatic environment, NPs undergo different classes of transformation, namely physical (aggregation, agglomeration, sedimentation, and deposition), chemical (dissolution, photochemical reactions, oxidation, sulfidation, etc.), and biological (biodegradation and biotransformation) [27].

The colloid science principles that describe the agglomeration of NPs under various conditions were discussed in many studies [28,29]. The fundamentals of NPs’ agglomeration, classic and extended DLVO theories, and aggregation kinetic modeling are carefully overviewed in the book of Zhang (2014) [30]. The factors affecting the agglomeration processes were also thoroughly described in the work of Abbas et al. (2020) [31]. The important role among these factors is the combination of physical and chemical properties of CNMs including size, surface charge, surface functionalization, etc., [32,33], pH and ionic strength of the water, which affect the stability of particle suspension [34,35], and the interaction with natural organic matter (NOM), which can change the surface properties and block the pores trapping contaminants within these pore spaces [25,36]. Another important issue affecting the behavior of NMs in biological media is the absorption of proteins and the formation of the so-called “protein corona” which further modifies the particle properties [37]. The impact of “biomolecular corona” on the biodegradation of CNMs, and the recommendations and guidelines for future studies, were provided in a recent review by Mokhtari-Farsani et al. (2022) [38]. Considering the diversity of the factors described above, it is very difficult to develop principles and models to predict the behavior of NMs in aquatic media.

Based on the data of homo- and heteroaggregation, and sedimentation of NPs and natural colloids, Quik et al. (2014) demonstrated the application of the detailed Smoluchowski—Stokes model to describe the fate of NPs in natural waters [39]. The model of Markus et al. (2015) [40] modified previously developed models [41,42] which allows to predict the behavior of NPs in extended variations of circumstances, such as the variation of concentrations of suspended particulate matter. The main physical transformations of CNMs refer to particle size, porosity changes, and to the interaction with NOM and other particles dispersed in the medium [25].

Chemical transformation was mainly studied for metal oxide [43] and metal NPs, in particular for silver NPs [44,45]. The chemical transformation of CNMs is related to the surface reactions, which can change their surface properties [25]. It was shown that the favorite sites for the chemical transformation of CNMs are the edges of graphene sheets and the areas of defect or metal catalyst localization [46,47]. The process of CNMs’ transformation in aquatic media might be induced by environmental factors such as light irradiation, the presence of oxidants, reductants, or NOM [25]. Ultraviolet light can induce photodegradation of CNMs because of its high energy which can excite electron transition, inducing the production of reactive oxygen species and electron-holes [10,48]. Moreover, it was demonstrated that the presence of other chemicals increases the rate of photodegradation of CNMs [49]. The oxidation of CNMs under the impact of O_3_ and H_2_O_2_ was reported, and the reduction was produced by S^2−^ [50]. However, it should be noted that in the environment, CNMs form complexes with other particles, NOM, heavy metals, or other pollutants which made the transformation and degradation processes even more difficult [10]. The interaction of NMs with other substances in an aquatic environment was discussed in many studies [51,52], and in this regard, the ability of CNMs to be applied as absorbents and catalysts for the remediation of organic and inorganic pollutants [12] represents significant interest.

The following discussion will be focused on the (1) structure and properties, (2) colloidal behavior, and (3) physical and chemical transformation of the most applied types of CNMs [53,54,55], namely graphene, carbon nanotubes (CNTs), fullerene (C60), and carbon quantum dots (CQDs) in the aquatic environment.

### 2.2. Graphene and Graphene-Related Materials

Graphene is a two-dimensional sheet of sp^2^-hybridized carbon atoms which have a structure of a honeycomb lattice of six-membered rings. Graphene sheets can be considered as a basic form for the other carbonaceous materials. It can be stacked to a 3D graphite, rolled to 1D nanotubes, and wrapped to 0D fullerenes [56]. The other common graphene-related materials represented by the oxidized form of graphene are named graphene oxide (GO) and reduced graphene oxide (rGO), produced by the elimination of oxygen-containing functional groups of GO.

The review work of Ren et al. (2018) [57] discussed in detail the factors affecting the colloidal behavior of graphene, GO, and rGO. These factors include physicochemical characteristics of materials and environmental conditions, such as the pH, ionic strength, salt type, and the presence of NOM, natural colloidal particles, and toxic heavy metal ions.

In general, graphene sheets have a strong tendency to form agglomerates and rebuild graphite [58,59]. The concentration and size of graphene play a crucial role in agglomeration. It was reported that in deionized water, few-layer graphene was rapidly agglomerated at concentrations of >3 mg/L [59]. Graphene nanoparticles with smaller lateral sizes tend to agglomerate more slowly and represent a higher risk for aquatic species [59]. Moreover, it was demonstrated that in saline water graphene has a high sedimentation rate and, interestingly, temperature had a higher impact on graphene surface tension than nanoparticle concentration [60].

The oxygenated surface of GO sheets allows them to form relatively stable suspensions, but in an aquatic environment, GO sheets intensively have contact with natural ions, NOM, colloidal particles, etc. In this case, the heteroaggregation of GO sheets with natural colloidal particles have more environmental importance compared to the homoaggregation of GO sheets [57]. As a result of heteroaggregation, the mobility of GO in an aquatic environment will be reduced. Most of the laboratory studies of GO heteroaggregation tested simple systems [61,62], while natural systems contain a variety of colloids usually associated with each other. The existing study of GO aggregation in a complex system (clay mineral kaolinite and metal hydroxide goethite) revealed that an increase in goethite concentration in kaolinite-goethite associations decreases the stability of GO, and heteroaggregation was strongly dependent on GO concentration, ionic strength, and pH of the system [62]. However, multiple studies showed that NOM can significantly improve the stability of GO in water due to steric repulsion [63,64]. It was highlighted that pH values of the natural environment, which vary between 5 and 9, will have no effect on the fate of GO by themselves [57]. In the research work of Adeleye et al. (2020), the authors demonstrated that GO agglomeration efficiency reached maximum, with a salinity of 1.33‰, and the highest sedimentation rate was at a salinity of 10‰ [65]. It was reported that CaCl_2_ destabilized GO more aggressively than MgCl_2_ and NaCl due to the binding capacity of Ca^2+^ ions with hydroxyl and carbonyl functional groups of GO [63].

Degradation of graphene-related materials in an aquatic environment occurs under light irradiation, oxidation/reduction processes, and after the contact with aquatic organisms and plants. Remarkably, the photolysis of graphene-related materials will undergo oxidation or reduction depending on the state of the carbon surface (oxidized or not) [57]. It was reported that graphene was oxygenated under visible light and formed new C-O-C and C-OH functional groups under the action of oxygen radical HO^•^, and became more stable and less toxic [58]. At the same time, GO is mainly transformed by photo-reduction under UV light exposure [66]. According to existing literature, the mechanism of GO reduction starts with water ionization and the formation of the HO^•^ and H^•^ radicals, and formation of solvated electrons (e^−^) [48,67]. After long-term irradiation, GO converts into CO_2_, low molecular-weight species, and products similar to rGO [48].

### 2.3. Carbon Nanotubes

Carbon nanotubes are considered to be the most promising and widely studied allotrope of carbon having a sp^2^-hybridized form and represent a graphene sheet rolled-up in a form of tube [68]. CNTs can be single-walled (SWCNT), double-walled (DWCNT), or multi-walled (MWCNT) depending on the number of concentric cylindrical layers [68]. Typically, the nanotubes have a very narrow diameter (nanometer-sized) and a relatively big length (micrometer-sized) [69]. CNTs represent either semiconductor or metal properties depending on the nature of their helix.

The colloidal behavior, and subsequent bioavailability and ecotoxicity, of CNTs in aquatic media is influenced by the physicochemical properties of NPs and the parameters of the media [70]. The hydrophilic nature and large particle length determine the poor stability of pristine CNT dispersions in water [71]. However, contact with NOM, liquid hydrocarbons, polycyclic aromatic compounds, or intentional surface modifications such as –OH and –COOH functionalities will reduce the hydrophobicity and increase the stability of CNT in water [72,73]. Schwyzer et al. (2011) in their work concluded that the fate of CNTs in an aquatic environment substantially depends on the state of CNTs at the moment of release [74]. In detail, “dry CNTs” demonstrated very low stability and most of the NPs were rapidly sedimented with no significant difference depending on medium conditions. On the other hand, “pre-dispersed CNTs” can be observed both in the water column and in the sediments, and in this case, the dispersibility of CNTs depends on stock suspension and media conditions. In the work of Schwyzer et al. (2013), the authors demonstrated that CNTs form spheroidal, bundle, and net-like agglomerate structures in a varying calcium-containing media, where calcium reduced the colloidal stability of CNTs by the neutralization of the particle charge and by acting as bridging agent [75]. Glomstad et al. (2018) also demonstrated that functionalized CNTs were more sensitive to changes in media properties than non-functionalized CNTs, and CNT surface oxygen content determined the dispersibility of CNTs in synthetic media but had no influence on CNT dispersibility in natural water [70]. Therefore, the dispersion, quantification, and characterization of CNTs in an aquatic environment is a very challenging task which depend on the multiple combinations of CNTs physical and chemical properties and media conditions. In this regard, the discussion about the development of standard protocols for the assessment of CNTs’ behavior in aquatic media becomes very important [76].

The chemical oxidation of CNTs requires strong oxidative forces and normally cannot be observed in the natural environment. However, the oxidation of CNTs can occur during the preparation of CNTs’ dispersions with ultrasonication [77]. It was demonstrated that carboxylated CNTs are able to intensify production of ROS, such as singlet oxygen (^1^O_2_), superoxide anion (O_2_^•−^), and hydroxyl radicals (^•^OH) under light irradiation, compared to unfunctionalized CNTs, which have no impact on ROS production [78,79]. The authors of the other study demonstrated that oxidized MWCNTs after the exposure to 256 nm UV light undergo photodecarboxylation and when a sufficient number of carboxylic acid groups have been removed, CNTs were rapidly aggregated [80]. Qu et al. (2013) demonstrated that carboxylated MWCNTs undergo significant photochemical transformation under UV irradiation, and the intensity of the transformation and the colloidal stability reduction were similar to sunlight exposure [81].

### 2.4. Fullerenes

Fullerene C_60_ is a molecule that has the shape of a soccer ball and consists of 60 carbon atoms, arranged as 12 pentagons and 20 hexagons. Each carbon atom is bonded to three others and is sp^2^-hybridized. C_60_ behaves like electron-deficient alkenes and reacts readily with electron-rich species [82]. The aqueous solubility of C_60_ is extremely limited by very high hydrophobicity [83].

Typically, unmodified fullerene C_60_ has a hydrophobic nature [84]. Although, the colloidal stability of C_60_ usually is reached by a specific preparation with the addition of organic solvents, such as benzene, acetone, toluene, and ethanol, with further dissolution in water, and distillation to remove most of the organic solvents; this procedure is called “solvent exchange method” [85]. The modified method of solvent exchange is currently used for the synthesis of C_60_ NPs [86]. The other possible methods of C_60_ dispersion in water include prolonged stirring, ultrasonication, and surface functionalization [87].

The used solvents and a chosen synthesis method influence the geometry and size of suspended C_60_ NPs which consequently affect the colloidal behavior and ecotoxicity of the NPs [85]. The aggregation kinetic of C_60_ was studied in several works and the critical coagulation concentration (CCC) was registered at around 85–150 mM NaCl and 4.1–4.8 mM CaCl_2_ at pH 5–6 [88,89]. The other study demonstrated that the CCC of C_60_ NPs produced by extended stirring in water was 166 mM KCl, compared to C_60_ NPs produced by the solvent exchange with toluene which revealed CCC at 40 mM KCl, the pH value of both experiments was 5.5 [90].

Chen et al. (2010) assumed that very good agreement of experimentally obtained C_60_ stability ratios with theoretical DLVO predictions demonstrate that the charges are consistently distributed on the C_60_ nanoparticle surface [85], because it is the main simplifying assumption of DLVO theory [91].

The presence and content of NOM may play a critical role in the transport and toxicity of C_60_ in the natural aqueous environment. Xie et al. (2008) showed that the addition of NOM caused concentration-dependent disaggregation of C_60_ crystals in natural water [92]. The work of Mashayekhi et al. (2012) revealed that NOM can promote or inhibit aggregation of C_60_ NPs depending on the properties and structural characteristics of different NOMs [93]. It was shown that more hydrophobic NOMs had a higher affinity to C_60_ aggregates, and that long chain molecules of NOM can cause precipitation of nanoparticles from water.

In the environment, C_60_ seems to be comparatively stable. Under UV light irradiation, C_60_ can undergo oxidation and polymerization [94]. It was reported that oxidized C_60_ (O–C_60_, epoxides) can be effectively degraded in water by the photo-Fenton method to give CO_2_, H_2_O, and a few of organic molecules [95]. In their work, the authors demonstrated the method of O–C_60_ removal from the water with the addition of FeCl_3_ and H_2_O_2_, and 72 h of UV light irradiation (185 nm). Moreover, the authors comprehensively described the degradation pathway of O–C_60_ by the photo-Fenton reaction. Sanchís et al. (2018) studied the impact of salinity, humic acids, and pH with or without light irradiation; they demonstrated that light stimulated the production of O–C_60_ and caused its further elimination after 24 h [96]. Therefore, heteroaggregation and the presence of particulate matter in the water column can limit the access of light and reduce oxidation plus further photodegradation of C_60_.

### 2.5. Carbon Quantum Dots

Carbon quantum dots (CQDs) or fluorescent carbon NPs are a relatively new class of carbonaceous nanomaterials created as an environmentally friendly and cheaper alternative to semiconductor quantum dots which usually have high toxicity due to the use of heavy metals in their production [97]. CQDs are typically quasi-spherical 0D carbon nanostructures with sizes below 10 nm [98]. CQDs have amorphous nanocrystalline cores with predominantly graphitic or turbostratic carbon (sp^2^-hybridized) or graphene and graphene oxide sheets fused by diamond-like sp^3^-hybridized carbon insertions [97]. The content of carboxyl moieties in the oxidized CQDs ranges from 5 to 50% (weight) [99]; these facilitate excellent water solubility and access further functionalization and surface passivation [100]. The function groups on the surface of CQDs can be represented by amino groups, oxygen, and polymer chains [101]. The function group composition significantly affects photoluminescence activity, the energy gap, and the energy level of the surface [102].

CQDs demonstrate highly hydrophilic properties and cell permeation [98]. Unlike graphene-related materials, which tend to aggregate in the media with high ionic strength, the unique properties of ultrasmall CQDs made them soluble and highly mobile in a variety of water conditions [103]. The colloidal behavior of CQDs in aquatic environments has been poorly studied to date. Bayati et al. (2018) for the first time investigated the colloidal stability of CQDs and confirmed a very high stability of suspended CQDs [104]. The authors assumed that CQDs will remain stable in most natural freshwater bodies but may agglomerate in seawater and groundwater conditions. Moreover, the study did not reveal direct correlation between particle size and water chemistry (ionic strength, pH, and NOM content). However, it was showed that the aggregation of amino-functionalized CQDs can be further inhibited in the presence of humic acid. For plain CQDs, the same effect was observed only in presence of both humic acid and CaCl_2_.

Due to particle size, CQDs have a redox capacity similar to photocatalysis and are often applied for organic pollutant degradation and water sterilization [105]. The other possible mechanism of pollutant degradation by CQDs is photooxidation with production of O_2_^•–^ and ^•^OH [106]. Despite CQDs being considered as low toxic materials, they can cause a threat to aquatic species as demonstrated in a few studies [107,108,109]. Moreover, CQDs are an effective antimicrobial agent [110]. Liu et al. (2021) in their work showed that CQDs undergo photodegradation in water and they investigated the effects of particle size, light intensity, light wavelength, temperature, pH, and ionic strength on the photodegradation kinetics of the laboratory-synthesized CQDs [111]. The authors also demonstrated that degradation products formed after CQDs’ irradiation with white fluorescent light had cytotoxicity in normal and malignant human cells.

## 3. Overview of Carbon-Based Nanomaterials Biodegradation

### 3.1. Enzymatic Biodegradation

The biological transformation of NPs is the most environmentally friendly degradation method [112]. Biodegradation of CNMs occurs due to the interaction of NPs with enzymes, organisms, and individual cells [113]. This section will discuss the main principles and current achievements in the field of CNMs’ biodegradation.

The contact of enzymes with NPs results in the changes in both sides of this interaction. NPs can be modified, degraded, or synthesized, while enzymes can be immobilized and optimized for further application. It was reported that different types of CNMs can cause either inhibition or enhancement of enzyme activity [114]. The effect of NMs on enzymatic activity depends on the types of enzymes, environmental conditions, physical and chemical properties of NPs, intentional and environmental modification of NPs [115,116,117].

Among the enzyme properties, the amino acid composition and orientation of an enzyme’s 3D structures play the key role [118]. Therefore, the effective enzyme interaction with NPs can be obtained by a proper match between enzyme composition/orientation and NPs properties. Suitable environmental conditions (pH, temperature, ionic strength, etc.) are also required for effective nano-bio interaction.

Enzymatic transformation of CNMs is the most studied and promising approach that can be used both for environmental purification from CNMs [119,120] and green synthesis of CNMs [121]. One of the first studies of enzyme-catalyzed degradation of CNM was performed by Allen et al. (2008) with SWCNTs and the plant enzyme horseradish peroxidase (HRP) [122]. In their following work, the same research group suggested that oxidation and degradation of CNTs occurs due to the hydrophilic interaction between the heme active site of HRP and the oxygen-containing defective sites on CNTs [123]. Zhao et al. (2011) showed the layer-by-layer degradation mechanism of MWCNTs exposed to HRP and H_2_O_2_, and highlighted that side wall defects facilitated degradation efficiency [124]. Kagan et al. (2010) demonstrated that bio-peroxidases, such as the human neutrophil enzyme myeloperoxidase (MPO) and its reactive radical intermediates, can catalyze the biodegradation of SWCNTs in vivo [125].

In general, mechanisms and achievements in enzymatic biodegradation of NPs, and the involvement in this process of reactive intermediates such as oxo-ferryl iron, hypochlorous and hypobromous acids were reviewed in a comprehensive work of Vlasova et al. (2016) [119]. Currently, the list of enzymes involved in the degradation of CNMs includes HRP, myeloperoxidase (MPO), lactoperoxidase (LPO), manganese peroxidase (MnP), lignin peroxidase (LiP), eosinophil peroxidase (EPO), xanthine oxidase (XO), and others [10,119]. Sureshbabu et al. (2015) demonstrated the importance of surface modification of NPs in their enzymatic degradation [126]. Moreover, metal-containing CNMs revealed peroxidase-like activity which can be responsible for self-biodegradation mechanism [119].

### 3.2. Microbial Biodegradation

Microorganisms have contact with all the substances present in the environment and take part in the production of a significant part of biomass on Earth. Chen et al. (2019) systematically reviewed the differences between the impact of varied types of CNMs on microbial communities [127]. It is worth noting that aside from negative effects, CNMs can stimulate the growth and the metabolic activities of tolerant microorganisms at relatively low concentrations. The ability of microorganisms to use CNMs as a source of carbon and the subsequent degradation of CNMs can be an accompanying effect of such microbial tolerance [128]. Therefore, microorganisms will inevitably have contact with CNMs [129]. The main mechanism of microbial transformation and degradation of CNMs is related to the ability of microorganisms to produce oxidative enzymes such as laccase, MnP, and LiP [130]. Microbial degradation of CNMs attracts the scientific community’s attention as a promising method of environment purification despite the current level of development of the process being fairly inefficient and not well-known [128].

Chen et al. (2017) summarized the works related to bacteria and fungi, which can degrade CNTs and graphene-related materials [112]. In Table 1, we gathered the results of research works in recent years that are related to application of microorganisms for CNMs’ degradation.

Aside from the transformation and degradation of CNMs, the contact of microorganisms with this type of NPs can have other beneficial effects, such as an alteration of biomass production in cultural plants [144,145], and the enhancement of organic pollutants’ degradation in the soil and the water environment [146,147].

Further studies are needed to improve the efficiency of microbial degradation of CNMs. These future works should include the development of mixed cultures of carbon-degrading microorganisms, identification of new species, and enzymes with high CNMs-degrading potential, exploring the impact of particle characteristics and media conditions on the process of biodegradation.

### 3.3. Biodegradation in Inflamatory Cells

The biomedical application of CNMs triggered the studies of their safety assessment including accumulation, excretion, impact on immune system, bio-corona formation, and degradation in higher organisms [148]. From this point of view, the role of the immune system which facilitates ‘digestion’ of CNMs through oxidative reactions represents a high interest in the elimination of possible toxicity of CNMs [148]. Moreover, the study of CNMs’ biodegradation in inflammatory cells was one of the most informative models that has helped to understand the main mechanisms of this process.

It was stated that the effective degradation of CNMs required a generation of reactive intermediates and a presence of a source of their oxidizing equivalents [119]. In this regard, inflammatory cells (neutrophils and eosinophils) support the production of the radicals (ROS and RNS) that are highly reactive towards CNMs [149,150] and facilitate enzymatic biodegradation of CNMs, specifically neutrophils via the production of MPO [125,151], and eosinophils via the production of EPO [152].

As reported by Kagan et al. (2010), neutrophils required activation to enhance the oxidative biodegradation of CNMs [125]. The other study demonstrated that neutrophils can produce so-called neutrophil extracellular traps (NETs) consisting of nuclear chromatin fibers studded with granule proteins which can capture and digest SWCNTs [148].

The ability of macrophages to digest CNTs was reported in several works [153,154]. Yang and Zhang (2019), in their work, reviewed the mechanisms and recent works related to the biodegradation of CNTs by macrophages in vitro and in vivo [155]. For macrophages, it was shown that the biodegradation of NPs occurred by the production of peroxynitrite (ONOO^−^) and the activation of the superoxide/peroxynitrite oxidative pathway [153]. The mechanism of peroxynitrite-driven oxidation is independent of the protein-nanoparticle binding and, therefore, can effectively oxidize pristine CNMs increasing their reactivity to the enzymes of neutrophils and eosinophils [153]. The other study demonstrated a strong dependence of SWNTs’ degradation by MPO and ONOO− from NADPH oxidase [156]. Lu et al. demonstrated that the binding of fibrinogen reduces the toxicity of SWCNTs but does not inhibit biodegradation via MPO and ONOO− dependent pathways [157].

Further studies of CNMs’ degradation and transformation mechanisms via cell degradation will allow to regulate the fate of carbonaceous NPs in vivo and facilitate the production of biodegradable-by-design NPs.

## 4. Biodegradation of Carbon-Based Nanomaterials in Aquatic Species

The discussion about the biological transformation and degradation of CNMs in the aquatic environment should implement the understanding of the processes represented in Section 2 and Section 3. In the other words, the physical and chemical transformation of CNMs released in water bodies may prevent their biological transformation by microorganisms, enzymes, or cells.

The emissions of carbon nano- and microparticles formed by forest fires, fossil fuel combustion, and other sources, which are called black carbon, have a significant impact on the environment [158,159]. Black carbon comprises 9% of the total organic carbon in aquatic sediments [160], and aquatic species had been influenced by black carbon for a very long period before manufactured CNMs were discovered [161,162]. Such an osculation suggests that aquatic organisms can have the mechanisms to reduce the toxicity of CNMs and possibly could transform and degrade CNMs.

Toxicity, uptake, and accumulation of CNMs were previously described in microalgae [163], *Daphnia magna* [164,165], bivalves [166,167], fish [168,169], and fish cell lines [170,171]. The other aspect includes the food chain transfer of CNMs, as it was reported in several studies, for the systems including bacteria, microalgae, crustacean, and fish [172,173,174].

Zaytseva and Neumann (2018) overviewed the penetration and accumulation of CNMs in terrestrial and aquatic plants with further analysis of the possible implications of CNMs in food chains [175]. The reduction of GO was demonstrated after contact with carrot root under the presence of endophytic microorganisms [176]. Moreover, the reduction of GO was reported with *Escherichia coli* [177]. The other study with the bacteria *Shewanella oneidensis* MR-1 demonstrated that the reduction of GO was catalyzed by the Mtr respiratory pathway, which facilitates the transfer of electrons from the interior of the cell to the external terminal electron acceptors [178].

The case study with the green algae *Desmodesmus subspicatus* and MWCNTs demonstrated the bioavailability of CNMs to the microalgae which was most probably supported by the sorption of extracellular polymeric substances produced by algal cells or other NOM presented in the media [163]. Moreover, this study demonstrated that 55% of absorbed CNTs were eliminated within three days. Despite the significant number of comprehensive toxicity studies related to the influence of CNMs on microalgae [179,180], to the best of the author’s knowledge, the problem of CNMs’ transformation after such an exposure remains out of the scope of the research.

The other study demonstrated a high accumulation of CNTs in the marine bivalve *Mytilus galloprovincialis* with minimal toxic damage to the mantle, gills, or digestive tract [181]. The mussels in this study excreted >110 mg CNTs g^−1^ while a higher accumulated concentration of CNTs observed in viscera was approximately 1 mg CNTs g^−1^. Therefore, bivalves can filtrate and concentrate CNMs in biodeposits.

The ability of mussels to adhere to various surfaces by using so-called mussel adhesive proteins (MAPs), and the fact that dopamine could mimic the function of MAPs and form polydopamine coating, build the area of mussel-inspired chemistry with various applications [182,183]. Alongside those applications is the surface modification of CNMs. This technique was applied for surface functionalization of GO [184] and CNTs [185,186] with the prospect of modification of the other types of CNMs. These applications illustrate that aquatic species had different unexplored mechanisms which can be used for CNMs’ transformation and degradation either directly or after several modifications.

The accumulation of MWCNTs was demonstrated in zebrafish (*Danio rerio*), where the nanomaterials were mainly accumulated in the gut of all fish, but also were detected in the blood and muscle tissue [187]. The authors demonstrated a 10-fold reduction of CNTs’ uptake in the presence of NOM after 48 h compared to the experiment without NOM. The other study demonstrated that the accumulation of few-layer graphene in zebrafish depends on the size of NPs and the presence of NOM, namely larger particles had a 170-fold greater uptake, and the presence of NOM increased the accumulation of graphene in zebrafish [188]. The study of Lammel and Navas (2014) showed low cytotoxicity of GO and carboxyl graphene in the fish cell line PLHC-1 in vitro with a high production of ROS [170]. The results of this study demonstrated that fish cells have a defense mechanism against CNMs and required further study.

## 5. Conclusive Remarks

The wide application of CNMs in pollutant removal, drug delivery, and other aspects of everyday life made safety management one of the primary problems. High stability of unmodified CNMs in water became a factor increasing the risks for the environment and living species. On the other hand, the contact of aquatic species with carbon-based materials is an interesting field of study considering the potential of aquatic species for pollution adaptation and further environmental remediation.

Except for the benefits to environmental purification, the understanding of the process of CNMs’ transformation and biodegradation will facilitate the development of “biodegradable-by-design” NPs, which also promise new applications in the area of nano-bio interaction, such as medicine and consumer products. Further study of the biodegradation of CNMs will allow to regulate the lifetime of NPs, which is especially important for drug nano-carriers and for reducing the toxicity of CNMs after involuntary exposure.

## Figures and Tables

**Table 1 nanomaterials-12-04149-t001:** The current studies of microbial degradation of CNMs.

Species	Types of CNMs	Results	Ref.
**Bacteria**			
*Labrys* sp. WJW	GO	isolation and identification of a novel bacterium which can degrade and use GO as a sole carbon source; systematization of GO derivatives and up-regulated proteins potentially responsible for GO degradations (oxidoreductases, lyases and hydrolases)	[131]
	GO, rGO, SWCNT, and o-SWCNT	significant influence of CNM characteristics on biotransformation; revealing of aerobic biotransformation mechanism via Fenton-like reaction	[132]
	MWCNT	degradation by loss and change of fibrillary structures in functional groups of MWCNTs; defects in basic structure; aerobic biotransformation via Fenton-like reaction	[133]
*Mycobacterium vanbaalenii PYR-1*	MWCNT, c-MWCNT	higher degradation of c-MWCNTs compared to pristine MWCNTs; potential mineralization of MWCNTs	[134]
Bacteria isolated from graphite mine	GO, rGO	oxidation of graphitic materials; higher oxidation of rGO compared to graphite; formation of holes in GO	[135]
Bacterial community of *Burkholderia kururiensis*, *Delftia acidovorans*, and *Stenotrophomonas maltophilia*	MWCNT	degradation of MWCNTs in the pretense of an external carbon source; identification of intermediate products;	[136]
**Fungi**			
*Phanerochaete chrysosporium*	rGO	increase of defects on carbon skeleton; oxidation of rGO; enzymatic oxidation prevailed on the impact of Fenton systems; higher transformation of NPs wrapped in the fungal balls	[137]
	MWCNT, o-MWCNT	both types of NPS were oxidized and shortened; precipitated o-MWCNTs showed more short tubes; defects on carbon skeleton; laccase and MnP were responsible for the transformation	[138]
	CQDs	CQDs did not affect he Lac and MnP activities, and did not induced oxidative damage; the decomposition activity of *P. chrysosporium* kept unchanged	[139]
*Trichoderma* sp. WF29, *Irpex lacteus* WF36, and *Trametes versicolor*	MWCNT	identification of size, surface charge, and pH change; measurement of involved enzymes (laccase, MnP, LiP)	[130]
*Cladosporium* sp.	Graphene, GO, SWCNT	all the tested NPS increased production of laccase, MnP, and LiP with the highest effect on MnP; CNMs acted as adsorbent of extracellular enzymes (especially SWCNTs) and as electron conductors; CNMs can increase lignin consumption by fungi	[140]
*Trametes versicolor*	SWCNT	no significant degradation of pristine SWNT was observed over six months	[141]
*Trametes versicolor* and *Phlebia tremellosa*	SWCNT, c-SWCNT	metal catalyst-rich and c-SWCNT promoted significant changes in the activity of peroxidase and laccase enzymes compared to pristine SWCNTs	[142]
**Mixed cultures**			
Soil microbial microcosm	C_60_ fullerene, C_60_ fullerol	intense fullerol mineralization compared to pristine fullerene	[143]
Soil microbial microcosm	C_60_ fullerene	report of the coupled process of photochemical and microbial transformation of C_60_; no increase of laccase or peroxidase enzyme activity; very low rate of C_60_ mineralization	[144]

o-SWCNT, oxygenated single walled carbon nanotube; c-SWCNT, carboxyl functionalized single-walled carbon nanotube; c-MWCNT, carboxyl functionalized multiwalled carbon nanotube; MnP, manganese peroxidase; LiP, lignin peroxidase.

## Data Availability

Not applicable.

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
