# Peer review of "Colloidal Behavior and Biodegradation of Engineered Carbon-Based Nanomaterials in Aquatic Environment"

_nanomaterials, 2022, doi:10.3390/nano12234149_

Round 1
Reviewer 1 Report
Knowledge on transformation and ultimate fate of nanoparticles is of the primary importance in the World, in which people begun to design, produce and apply continuously growing amounts and diversity of classes of the aforementioned substances. Because these substances may pose certain danger to various forms of life they definitely deserve special attention. Authors prepared a review paper on behavior of graphene, graphene oxide, carbon nanotubes, multi-walled carbon nanotubes, fullerenes and quantum dots in aquatic environment. Particularly there are presented recent achievements related to fate of these nanoparticles in all kinds of living organisms as well as on their (bio)degradation. The manuscript is well organized and clearly written. Its content is based mainly on papers published mainly in the last decade. However, I see a problem with the title. Namely, the expression “carbon-based substances” (materials, chemistry) is commonly reserved for the much broader class, i.e. for organic compounds (organic chemistry. Thus, with purpose to avoid disappointment of some readers I propose to list classes of discussed compounds by their names.
Author Response
Dear Editor,
Thank you for the valuable comments. We revised the manuscript and corrected the title. Namely, we have specified the class of nanomaterials as ‘enginered carbon-based nanomaterials’ which is the most convenient name of this group and it is correlate whith the literature.
Reviewer 2 Report
The behavior of typical carbon-based nanomaterials in a water environment is summarized in this publication. The authors focus on their colloidal behavior in the aqueous environment and the potential for biodegradation after a brief introduction to the uses of graphene, carbon nanotubes, fullerenes, and carbon quantum dots. This type of review is challenging due to the extensive broadness of the topic. I have a few suggestions below that might help improve the presentation.
Comments 1: The amount of defects and oxidation in graphene sheets, the change in charge in various pH solutions, and the modification of various functional groups on the surface of graphene, in my opinion, all have a significant impact on the material's stability and degradation in a water environment, but the author has not addressed these issues.
Comments 2: The destiny of carbon-based materials in the environment has been extensively debated. I wish the author could present a more creative argument. For instance, carbon-based compounds are frequently employed in water treatment because of their ability to adsorb environmental contaminants. When mixed with environmental pollutants, what happens to them in the water environment?
Comments 3: What is the current, widely used process for degrading carbon-based materials in the aquatic environment? What about the author's mention of a practical application for biodegradation? Which advantages does it have?
Comments 4: Why does the author not discuss quantum dot biodegradation in Table 1?
Comments 5: Why is the inflammatory cell degeneration mentioned in the article on the fate of carbon-based materials in the aqueous environment? Is it relevant to the subject?
Comments 6: The fate of carbon-based materials is the subject of the article's title. In reality, there are several behaviors in addition to dispersion and agglomeration, including deposition, bioaccumulation, nutrient transformation, etc. It is questionable whether the title of this article is appropriate because the author did not go into detail about it.
Round 2
Reviewer 2 Report
No other comment